# Suboptimal MMR Vaccination Coverages—A Constant Challenge for Measles Elimination in Romania

**DOI:** 10.3390/vaccines12010107

**Published:** 2024-01-22

**Authors:** Aurora Stanescu, Simona Maria Ruta, Costin Cernescu, Adriana Pistol

**Affiliations:** 1Faculty of Medicine, Carol Davila University of Medicine and Pharmacy, 050474 Bucharest, Romania; aurora.stanescu@insp.gov.ro (A.S.); adriana.pistol@umfcd.ro (A.P.); 2National Institute of Public Health, National Centre for Communicable Diseases Surveillance and Control, 050463 Bucharest, Romania; 3Department of Viral Emerging Diseases, Stefan. S. Nicolau Institute of Virology, 030304 Bucharest, Romania; 4Romanian Academy, 010071 Bucharest, Romania; cernescucostineugen@gmail.com

**Keywords:** measles, vaccine coverage, outbreak, Romania, vaccine hesitancy

## Abstract

Measles is targeted for elimination since 2001, with a significant reduction in cases recorded worldwide, but outbreaks occur periodically due to immunization gaps. This study analyzes the evolution of vaccination coverage rates (VCRs) in Romania, a EU country with large measles epidemics during the last two decades, including an ongoing outbreak in 2023. Vaccination against measles has been part of the National Immunization Program since 1979, initially as a single dose, and from 1994 onwards it has had two doses. The initially high national VCRs of >97% gradually declined from 2010 onward and remained constantly under 90%, with further decreases during the COVID-19 pandemic. The lowest VCRs for both vaccine doses in the last decade were recorded in 2022 and were 83.4% for the first dose and 71.4% for the second dose, with significant differences among Romania’s 42 counties. Several factors contributed to this decline, including failure to attend the general practitioners’ offices, increased number of children lost to follow-up due to population movements, missed vaccination opportunities due to temporary medical contraindications, a surge in vaccine hesitancy/refusal, a decreasing number of general practitioners and discontinuities in vaccine supply. The persisting suboptimal VCRs in Romania threaten the progress toward measles elimination.

## 1. Introduction

Measles is an acute, self-limiting viral infection, with universal spread and high contagiousness that primarily affects children, and has potentially severe complications such as pneumonia, otitis, acute encephalitis, subacute sclerosing panencephalitis, prolonged anergy and death.

Measles elimination is a feasible target, if vaccination coverage rates (VCRs) of at least 95% with two doses of the highly effective live-attenuated measles-containing vaccine are achieved and maintained worldwide [1,2,3].

Important milestones in the WHO strategy toward measles and rubella elimination [4] have been achieved globally during the last two decades, with significant reductions in the burden of both diseases, improved surveillance, and increased vaccine uptake in the national immunization programs. Nevertheless, measles outbreaks continue to occur periodically, due to persisting immunization gaps in the population, and the disease remains one of the leading causes of death in children worldwide [5,6].

Romania is an EU country with over 19 million inhabitants [7] situated in south-eastern Europe, with a border with Ukraine, where the current armed conflict caused a large displacement of the population, influx of migrants in the neighboring countries and disruption of medical services, including immunization programs. 

Measles vaccination was introduced in the National Immunization Program (NIP) in Romania in 1979. From 1979 to 1993, a single dose of the monovalent measles vaccine was recommended for children aged 9–11 months. In 1994, a second measles vaccine dose was introduced for children aged 6–7 years of age (first school grade). The combined measles, mumps and rubella (MMR) vaccine replaced the monovalent vaccine in 2004, with a first dose (MMR1) administered at 12 months of age and a second dose (MMR2) administered at 6–7 years (age at school entry) from 2005 onwards. Since 2015, the recommended age for MMR2 was lowered to 5 years, to ensure vaccination before school admission. Although vaccination is not mandatory, it is highly recommended, and vaccines are provided free of charge to all children registered in the healthcare system. Supplementary immunization activities (SIA) were organized periodically to increase the vaccine coverage rates. The 1993 catch-up SIA for all children aged 1–14 years reached a VCR of 97% and the 2002 follow-up SIA for children aged 1–4 years attained a vaccine coverage of 87%.

### Measles Epidemiology in Romania 

In the pre-vaccination era, measles incidence rates in Romania were high (120–140 cases/100,000 inhabitants). Following the introduction of monovalent measles vaccination in the NIP, measles incidence rates declined to below 60 cases/100,000 inhabitants, and smaller outbreaks emerged at longer intervals compared to those in the pre-vaccination era [8]. After a national measles immunization campaign in 1998–1999 (targeting all children aged 7–18 years who lacked immunization records of 2 vaccine doses) [9] and the implementation of a national case-based surveillance in 1999, a further drop in measles incidence was recorded (0.3 cases per 100,000 inhabitants in 2000). Romania entered the measles elimination phase and 2008–2009 was the first period with measles notification rates of <1 case/million inhabitants. In 2010, there were 193 measles cases (a notification rate of 0.9%000), mostly due to a small outbreak in counties in the north and north-eastern regions [10].

Nevertheless, two large-scale measles epidemics emerged at short intervals of time during the last two decades (2011–2012 and 2016–2020). During the 2011–2012 outbreak, there were 12,234 confirmed cases of measles, including 3 deaths. Children <1 year of age, who were not eligible for MMR vaccination according to the national schedule, were the most affected group (incidence >700/100,000). The predominant measles viral strain identified belonged to genotype D4, which is endemic to Romania [11]. Between 2011 and 2012, Romania faced a concurrent epidemic of rubella, caused by genotype 2B, with 24,627 confirmed cases, without deaths [11]. During the 2016–2020 measles outbreak there were more than 20,000 confirmed cases and 64 deaths [11]. Again, children under 1 year were the most affected age group (incidence > 900/100,000), followed by those aged 1–5 years. One important factor in the transmission of the disease was non-compliance with isolation measures in hospitals/pediatric wards/health care facilities, resulting in nosocomial infections. The outbreak started in the north and north-western regions of the country, in communities whose members frequently migrate from one county to another, but also outside Romania, and tend to have low vaccination rates. The predominant measles virus strain belonged to an imported genotype, B3 [12]. The national surveillance system for measles and rubella did not detect rubella cases during the measles outbreak in 2016–2020. With this background, the present study aims to analyze the evolution of measles vaccine coverage rates and its impact on the measles elimination process in Romania.

## 2. Materials and Methods

Measles surveillance data and MMR vaccination coverages rates were extracted from the national database that includes all data collected by the National Centre for Communicable Diseases Surveillance and Control in Bucharest (NCDC). In Romania, measles is a notifiable disease since 1978, and suspected measles cases must be immediately reported to the local Public Health Authorities, which report to the NCDC. A national case-based notification for measles was initiated in 1999, and the European Union (EU) case definition and classification [13] have been adopted since 2005. Suspected measles cases are investigated using a case form with clinical and epidemiological data and laboratory testing of a blood specimen or nasopharyngeal swab. All reported measles cases are confirmed by the presence of IgM anti-measles antibodies or measles RNA detection by RT-PCR. Data are collected and analyzed at the NCDC and monthly reports are submitted to the European Center for Disease Prevention and Control (ECDC) through the electronic European surveillance system (TESSy) and to the World Health Organization.

Like most vaccinations included in the Romanian NIP, the MMR vaccination is carried out by general practitioners. Immunizations are both recorded on general practitioner’s office registry and in the national electronic vaccination registry (RENV), and vaccination certificates are issued for all vaccinated persons, including foreign citizens. Vaccination coverage rates for each vaccine dose are estimated by reviewing the immunization records submitted by general practitioners and using as a denominator the population of each county, as provided by the National Institute for Statistics. The information is collated and validated from each of the 42 districts in Romania and assessed by NCDC repeatedly for each year’s birth cohort at 12, 18, and 24 months of age. 

To assess the reasons for non-vaccinating, we performed a cross-sectional analysis of the responses received in a dedicated questionnaire (presented in the Appendix A), that is distributed regularly to all general practitioners in each of the country’s 42 districts. The questionnaire includes items on the number of vaccinated children for each antigen included in NIP; the reasons for non-vaccination including vaccination refusal; missed presentation; vaccine shortages (all by type of vaccine and number of doses). Data segregated by high-risk subpopulations are not available. The analysis included questionnaires distributed at the national level during the last measles outbreak (2016–2020) and during 2020–2022 to identify factors related to suboptimal vaccination uptake. The statistical analysis was made using Stata/MP 13.1 for Windows using a two-sample test of proportions using groups (Z), with *p* < 0.05 considered statistically significant; the corresponding 95% confidence intervals (95% CI) were estimated.

A joinpoint analysis was used to assess changes and trends in vaccine coverages rates (VCR) for both MMR first (MMR1) and second (MMR2) dose. We used the software joinpoint Regression Program, Version 5.0.2–May 2023; Statistical Methodology and Applications Branch, Surveillance Research Program, National Cancer Institute. Based on this technique, the annual percentage change (APC) and the corresponding 95% confidence intervals (95% CI) were estimated to quantify a change in trends; we selected vaccine coverage rates (VCR) as the dependent variable and year as the independent variable. We applied an Empirical Quantile method for which the test statistic and *p*-value are not available. 

## 3. Results

### 3.1. Evolution of Vaccine Coverage Rates

According to the national database, after the introduction of the combined measles–mumps–rubella vaccine (MMR); the national vaccine coverage rates from 2005 to 2008 reached 96–97% for both doses. However, the first joinpoint was detected in 2008; from 2008 to 2015, a significant decreasing trend was observed for MMR1 coverage rates with a negative annual decrease of −1.74%. The second joinpoint was detected in 2015 and onwards, and an important but still suboptimal recovery was observed; between 2015 to 2019 the APC was positive with an annual increase of 1.69%. After this short period, a steep decline occurred and a third joinpoint was detected in 2019; from 2019 to 2022, the APC was negative with an annual decrease of −2.22% (Figure 1).

From 2005 to 2012, the MMR2 coverage rates had a decreasing trend with a negative annual decrease of −0.69. The first joinpoint was detected in 2012; from 2012 to 2015, a dramatically decreased trend was observed for MMR2 coverage rates with a negative annual decrease of −6.74%. The second joinpoint was detected in 2015. From this point onwards, the APC was constantly negative with an annual decrease of −0.05% (Figure 2).

Comparing MMR vaccine coverage data from the pre-pandemic year (2019) with data from the subsequent pandemic years (2020 and 2021), we note a decrease in the VCRs with more than 2% in 2020 and 2021 for MMR1, as well as an additional drop of more than 0.5% in the already low MMR2 coverage. In 2022, the vaccine coverage rates reached the lowest level in the last decade with 83.4% for MMR1 (with a 6.6% decrease compared to 2019) and 71.4% for MMR2 (with a 4.4% decrease compared to 2019) (Figure 3).

### 3.2. Sub-National Vaccination Coverage Rates

Significant differences in the vaccine coverage rates are present among Romania’s 42 counties. During the last two years, there was a significant increase in the percentage of counties with vaccine coverage rates of less than 75% assessed in children aged 12 months, when the first dose of MMR is administered according to the national schedule, which is 16 out of 42 counties (38%) in 2021 vs. 31 out of 42 (73%) in 2022. Only three counties had a vaccine coverage rate higher than 90% each year and none have attained the 95% coverage required for the interruption of measles transmission (Table 1). However, an overall increase in VCR at age 18 months was recorded, based on results of our analysis (Table 1), with 35.7% of counties in 2021 and 21.4% in 2022 reaching vaccine coverage rates of more than 90% (vs. 7.1% at 12 months). Still, a vaccine coverage of at least 95% was reached in only 3 and 4 counties in 2021 and 2022, respectively.

### 3.3. Reasons for Non-Vaccination

Using a two-sample test of proportions with groups (Z), with *p* < 0.05 considered statistically significant and the corresponding 95% confidence intervals (95% CI) estimated, we performed an analysis of the questionnaires’ data collected from the 42 counties by the National Centre for Disease Control in the pre-pandemic years, which revealed the main reasons for non-vaccination (Table 2). Failure to attend the GP’s office was the most commonly stated reason and for which significant differences between urban and rural communities were present (45.9% in urban vs. 37.9% in rural communities, *p* = 0.001). This was followed by temporary medical contraindications, vaccination refusal (higher for MMR compared to other vaccines included in the national immunization program) and children lost to follow-up (born/living abroad), all without significant differences in urban vs. rural communities.

### 3.4. Measles Incidence 2020–2023

In 2020, 1004 measles cases were confirmed during the first 6 months of the year, despite all nonpharmacologic measures installed during the COVID-19 pandemic. Nevertheless, the incidence was 5.2 cases/100,000 inhabitants, and 4 times lower compared to 2019–2020 cases/100,000 inhabitants. Between 2021–2022, the number of reported measles cases was very low (with only 2 cases confirmed in 2021 and 10 in 2022), mirroring the general EU pattern. 

Yet, in 2023, 2735 measles cases were confirmed, including 3 deaths (as of 20 December 2023); 91% of all cases, including the fatal ones, were recorded in unvaccinated persons. Most measles cases (40.3%) in 2023 were among children aged 1–4 years old; the majority (94.9%) of these were unvaccinated. A proportion of 12% of the total number of cases diagnosed until now are in children <1 year of age; they are not eligible for MMR vaccination, according to the national immunization program (Figure 4).

As of 24 December 2023, the outbreak was concentrated mostly in central and western regions of Romania; the county most affected was that of Mures, with 748 cases (incidence of 144.49/100,000 inhabitants), followed by the neighboring counties of Brasov—657 cases (incidence 119.25/100,000 inhabitants) and Cluj—197 cases (incidence 28.7/100,000 inhabitants). In December 2023, the spread to the southern part of the country became apparent, with 352 cases reported in the capital city—Bucharest (an incidence of 20.3/100,000 inhabitants) and 123 cases in its suburbia in the neighboring Ilfov county (an incidence of 22.1/100,000 inhabitants). The increasing number of cases are also confirmed in the counties bordering Bulgaria (Giurgiu, Calarasi, and Teleorman). The 2022 vaccine coverage for both MMR1 and MMR2 in the most affected counties is presented in Table 3.

During the previous 5 years, counties with the highest number of cases tend to have constantly lower MMR1 coverage rates than the national average rate, both when assessed at 12 months and 18 months of age (Figure 5), with the exception of Cluj and Bucharest, which are the most populated counties in Romania (Bucharest—2.1 million; Cluj > 710,000 registered residents), and have a continuous influx of non-residents commuting from neighboring cities for work and a high number of national and international students.

## 4. Discussion

Measles remains endemic in Romania, due to constant suboptimal coverage rates with MMR vaccine doses. By analyzing the questionnaires collected periodically from general practitioners, our study demonstrates that the reversal of the progress made until 2010 has several reasons, including: (a) failure to attend the general practitioners’ office, due to either high migration rates with families moving frequently and children lost to follow-up or missing vaccination appointments, (b) heterogeneous and sometimes suboptimal caregivers’ information on the temporary contraindications for the vaccination and vaccination schedule, (c) a decreasing number of general practitioners, with dissimilar inter-county distribution, (d) a constant increase in vaccine hesitancy and widespread anti-vaccination movements and (e) discontinuities in the supplies of MMR vaccines. Further reductions in vaccination rates were recorded between 2020–2022, most probably due to movement restrictions and disruptions in the health system, as significant human resources were directed towards COVID-19 pandemic management. 

However, the most concerning factor is vaccine hesitancy, with an increasing mistrust regarding the safety of vaccination observed worldwide [14]. The widespread fear of unproved long-term adverse reactions following vaccination has become an important reason for not being vaccinated in Romania [15]. Accordingly, the country has one of the lowest COVID-19 vaccination rates in the European Union, with only 50.7% of the population vaccinated with a primary scheme and less than 10% with the first booster dose (compared to the overall rate of 82.4% and 54.8%, respectively, in EU/EEA). Moreover, the overall coverage rate for all vaccines is decreasing, as revealed by the latest data received by the NCDC [16]. In order to increase vaccine acceptance, more extensive and sustained communication campaigns are needed and a national communication strategy for vaccination must by implemented, with high-quality scientific information on the benefit of vaccination, as well as a correct and balanced view on the risk of adverse events following immunization [17].

Disparities in vaccine coverage rates at the sub-national level might be explained by the fact that the general practitioner’s network does not uniformly cover all of the country’s regions [18]. There are areas with scarce human resources and difficult access for the most vulnerable groups, such as unemployed persons with no health insurance, mono-parental families, people living in poverty and Roma minorities. During the last measles outbreak of 2016–2020, these groups were disproportionately affected by the disease, which clearly indicated inequities in immunization [15]. A recently published ethnographic study on the barriers and drivers towards vaccination in Romania identified disparities in clinics with low and high vaccine coverage rates (in terms of contraindication assessment, type of information provided to the child’s caregiver related to adverse events, openness to questions, appointment scheduling and correctitude of administration techniques) and defined interventions to increase the training of healthcare providers and to improve the services for the vulnerable and disadvantaged population [19]. A mid-term term solution could be the establishment of community centers, delivering integrated health care and prevention services, including vaccination and vaccine-related education. Integrated community centers can more efficiently serve the rural communities, ensuring a dedicated space for community nurses, as well as of socio-educational assistance services. This type of facility will ensure patients’ privacy and confidentiality and will favor teaming up with other specialists (doctors, other community nurses, midwives and health mediators). Nevertheless, support from the local authorities is necessary in order to establish health community assistance programs and to increase the access of vulnerable groups to social and medical services.

As reflected by the results of the present study, vaccination coverage rates increase at 18 months compared to 12 months of age. This indicates delays in vaccinations, allowing for the accumulation of susceptible children. Missing vaccination appointments, temporary contraindications for vaccination, vaccination hesitancy and dysfunctionalities in measles vaccine procurement and its distribution between counties can all be responsible for these delays, suggesting that a more detailed analysis at the local level is needed. In response to the current measles outbreak, supplementary immunization activities are ongoing, with catch-up vaccination campaigns being organized in each county and especially in communities with suboptimal vaccination coverage. Nonetheless, the results are modest, with only 10% of those aged between 1 and 18 years recovered for vaccination until October 2023. By mid-November 2023, the Ministry of Health officially declared a measles epidemic; an early-age vaccination campaign was initiated for children aged 9–11 months, who remain at high risk for measles as long as herd immunity is not ensured. 

Measles cases in adults remain infrequent, reflecting both a better vaccine coverage rate for those aged >18 years and the long-term persistence of immunity after natural infection (for older adults, born before the introduction of measles vaccination).

The constant decreasing measles-containing vaccine coverage in Romania explains the recurrent measles outbreaks observed during the last decade. As indicated by our study, during the ongoing 2023 outbreak, the highest attack rates are registered in counties with low vaccine coverage rates. The same observation was true for the previous outbreaks. For example, in 2015, using a mathematical model [20], a high-risk score was calculated for 13 counties (taking into account the vaccine coverages rates for MMR1 and MMR2, the number of measles cases during the previous three years and other contributing factors at the county level): these counties accounted for the highest number of cases during the 2016–2020 measles outbreak. It is well known that measles vaccination status is “the canary in the cold mine”, reflecting deficiencies in the overall performance of the immunization program. Indeed, the latest data from the NCDC [21] shows that the vaccination coverage is decreasing in Romania for almost all vaccines included in the national immunization program (inactivated polio vaccine, diphtheria–tetanus–pertussis vaccine, Hemophilus influenza vaccine, pneumococcal vaccine—all with vaccine coverage of 78.5% and hepatitis B vaccine—75.4% vaccine coverage). According to the last UNICEF report, in 2021, Romania had one of the highest shares of children with zero doses (9723; 5% shares of children under 1 year) and under-vaccinated children (17,510; 9% shares of children under 1 year) from countries in Europe [22].

A comparative analysis of measles vaccine coverage and recurrent outbreaks in Europe revealed interesting results. Following a resurgence of measles in the WHO European Region during 2016–2019, the number of reported measles cases declined significantly during May 2020–December 2022, coinciding with the COVID-19 pandemic [23]. This most probably reflected both the efficacy of nonpharmacological control measures, with a global low transmission of all respiratory viruses, and underreporting of the disease [24,25]. However, the COVID-19 pandemic has also caused a significant disruption of immunization programs worldwide, with a decrease in the number of children receiving the first dose of measles vaccine (from 86% in 2019 to 81% in 2021) [26]. In the WHO European Region, between 2010–2019, the measles VCRs increased from 94% to 96% for the first dose and from 80% to 92% for the second dose, but pockets of unimmunized populations were reported in all countries, comprising both children and adults [27]. In the European Union (EU) and the European Associated Area (EAA), small decreases in the VCRs were experienced in 2021 by 15 countries [28]. An ECDC Report from July 2023 signaled that already in 2018, apart from Romania, six other EU countries (Poland, Bulgaria, Estonia, Lithuania, Croatia and Cyprus) had suboptimal measles vaccine coverage rates (less than 89% for both vaccine doses) [29]. Although in 2022 the overall European regional MCV1 coverage was 93%, there are marked differences between countries in the region, with the lowest coverage rates with the first measles vaccine dose registered in middle-income countries in the Balkans, including a record low rate of 20% in Montenegro [27]. The European Immunization Agenda 2030 [2], launched at the end of 2021, aims to reduce these discrepancies in the vaccination coverage, addressing both the vaccine acceptance and vaccine procurement and logistics. These actions are urgently needed, as since the beginning of 2023, the WHO European Region has been experiencing an alarming increase in measles cases. Over 31,000 cases have been reported by 40 of the Region’s 53 Member States between January and December 2023 [30,31]. The highest numbers are registered in Kazakhstan (13,254 cases), the Russian Federation (6131 cases), Turkey (4602 cases) and Kyrgyzstan (3811 cases) [32]. In the EU/EAA, the total burden of measles cases was low during the first 10 months of 2023. However, there is a lag in reporting, as only 1453 cases of measles were registered, according to the last available ECDC report [33]. Out of these, 67.3% were in Romania, 10.6% were in Austria and 7.3% were in France. The epidemic situation remains stable in the other EU countries, with more than 30 cases reported only in Germany and Belgium. Nevertheless, during the last months of 2023, Poland and Italy have observed an increase in the number of cases [33], and enhanced surveillance is needed after the schools’ winter holidays.

Outbreaks might be further amplified by the high risk of measles importation of new measles strains, due to an increasing influx of labor immigrants from Asia and Africa, as well as refugees, particularly from Romania’s neighbor Ukraine, where vaccine coverage is also suboptimal, and nationwide measles outbreaks were recorded in 2017–2019 [34]. A previous study indicated that the measles virus strains identified during the current outbreak in Romania were phylogenetically related to Asian strains and different from the strain circulating during the last measles outbreak [35]. Since the beginning of war in Ukraine, the Romanian Ministry of Health and public health authorities have implemented, with WHO support [36], enhanced measles surveillance activities nationwide, along with a rapid implementation of catch-up vaccination strategies, especially in the counties bordering Ukraine. In addition, specific actions have been implemented to check the immunization status of all refugee children under 6 years of age, and free MMR vaccination is offered for immigrants, prioritizing children with missed doses in the first 14 days since entry [37]. Routine immunization activities and public information were intensified, with the distribution of multilingual health information booklets and leaflets beyond border crossing points, as well as postings on Facebook, and launching an official website in four languages. 

A national vaccination strategy and a specific plan for measles and rubella elimination were recently implemented in Romania, with specific measures aimed at ensuring a constant vaccine supply, an efficient and equitable vaccine distribution and increased accessibility [38]. A life-long vaccination approach was implemented for the first time, with MMR recommended and reimbursed for bone marrow transplant recipients and their family contacts who did not have specific detectable antibodies. Hopefully, taken together, these measures will sustain a progressive increase in vaccination uptake, which in turn will help reach the measles elimination target.

## 5. Conclusions

A suboptimal measles vaccination coverage persisted in Romania during the last decade, reversing the earlier progress toward measles elimination. The growing immunizations gaps in the population resulting in the accumulation of susceptible individuals has been the basis for several large outbreaks, including the ongoing one in 2023. An increasing vaccine hesitancy, amplified by the COVID-19 pandemic, as well as the increased population movements (with both Romanian emigrating and the increasing number of immigrants and refugees) represents a real threat for future outbreaks, with potential dissemination beyond Romania. 

## Figures and Tables

**Figure 1 vaccines-12-00107-f001:**
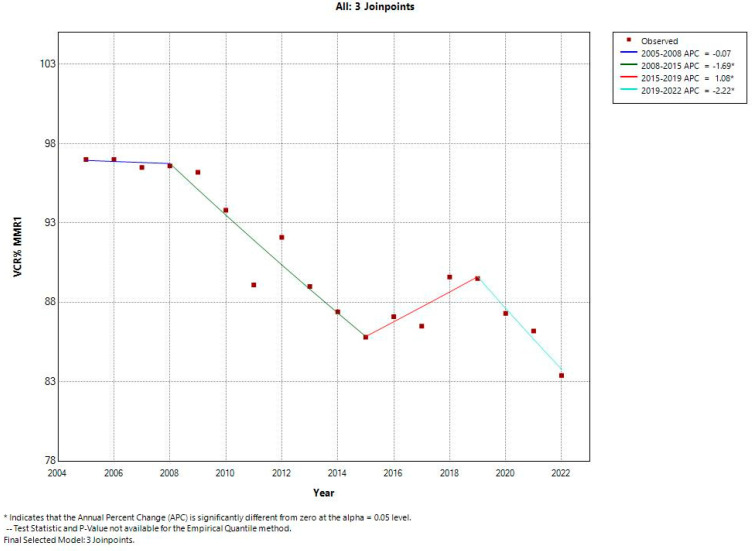
First MMR dose vaccine coverage rates trend (2005–2022): Romania.

**Figure 2 vaccines-12-00107-f002:**
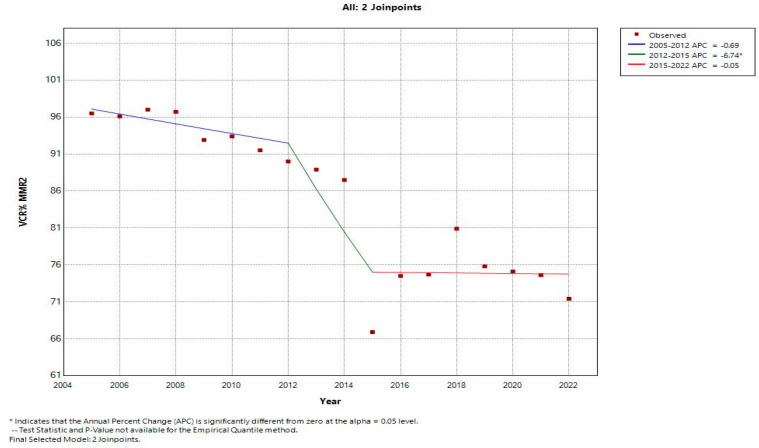
Second MMR dose vaccine coverage rates trend (2005–2022): Romania.

**Figure 3 vaccines-12-00107-f003:**
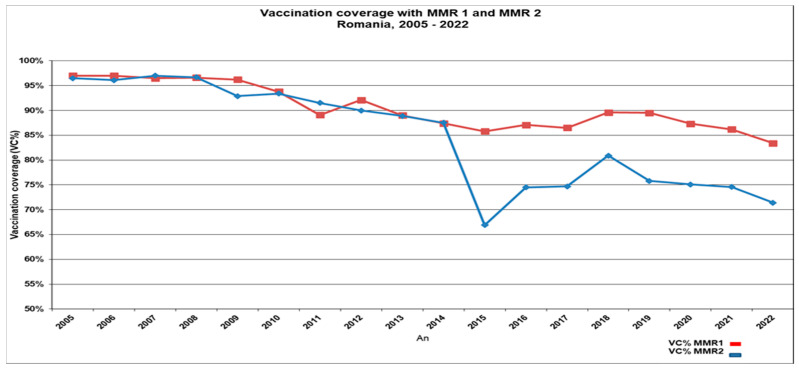
Vaccination coverage with the first and the second dose of MMR vaccine in Romania: 2005–2022.

**Figure 4 vaccines-12-00107-f004:**
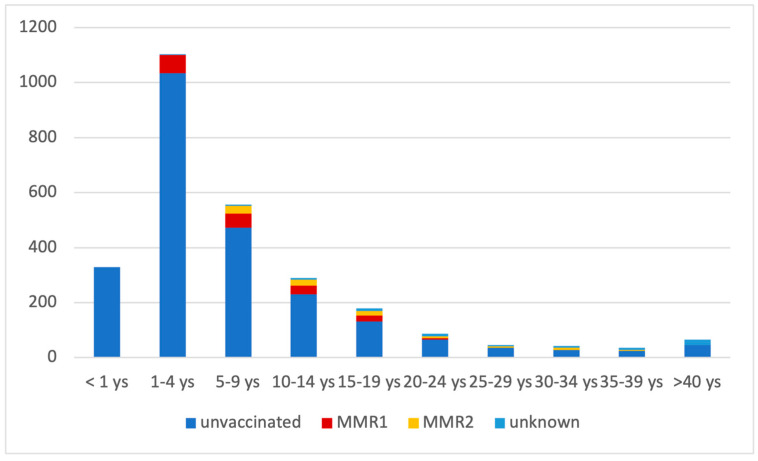
Age distribution of measles case in Romania, January–December 2023.

**Figure 5 vaccines-12-00107-f005:**
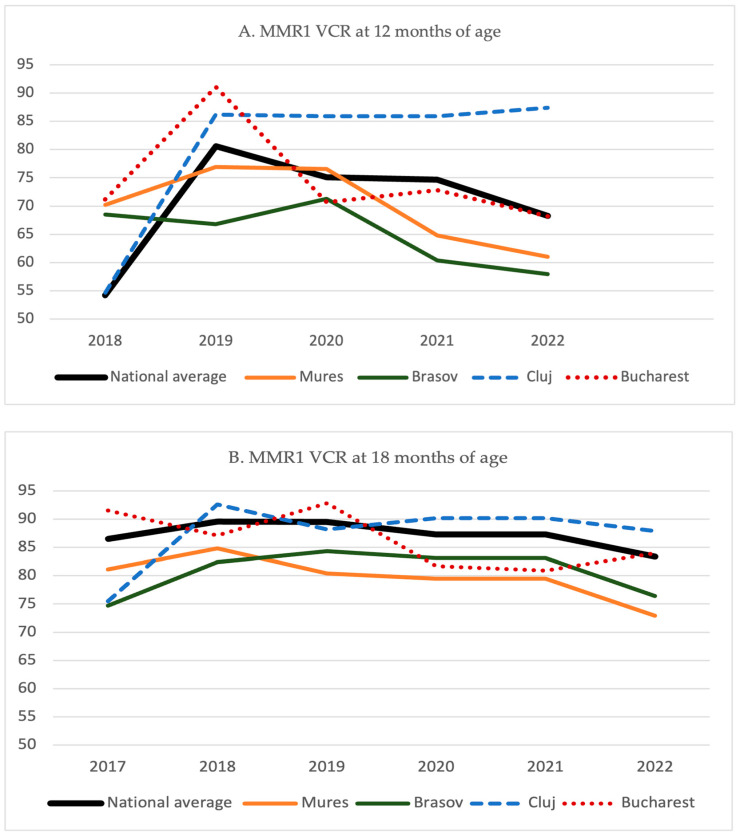
Evolution of the MMR1 vaccine coverage rates (reported at 12 months and 18 months) in the most affected counties during the 2023 measles outbreak, compared to the national average (in black).

**Table 1 vaccines-12-00107-t001:** Distribution of low and high vaccination coverage rates (VCR) assessed for children at 12 months of age and 18 months of age in the 42 counties of Romania.

	Counties with VCR < 75%		*p*	Counties with VCR ≥ 75%–≤ 90%		*p*	Counties with VCR > 90%		*p*	Counties with VCR > 95%		*p*
	N, (%)			N, (%)			N, (%)			N, (%)		
Age	12 Months	18 Months		12 Months	18 Months		12 Months	18 Months		12 Months	18 Months	
2019	8 (19%)	1 (2.3%)	0.001 (0.042; 0.295)	25 (59.5%)	14 (33.33%)	0.014 (0.059; 0.470)	7 (16.6%)	24 (57.1%)	0.000 (−0.592; −0.217)	2 (4.7%)	3 (7.1%)	0.679 (−0.051; −0.076)
2020	13 (30%)	1 (2.3%)	0.000 (0.133; 0.424)	25 (59.5%)	12 (28.6%)	0.004 (0.107; 0.510)	4 (9.5%)	22 (52.3%)	0.000 (−0.603; −0.252)	0	7(16.6%)	NA
2021	16 (38%)	2 (4.6%)	0.000 (0.174; 0.493)	23 (54.8%)	22 (52.3%)	0.818 (0.397; 0.698)	3 (7.1%)	15 (35.7%)	0.001 (−0.450; −0.121)	0	3 (7.1%)	NA
2022	31(73%)	9 (21.4%)	0.000 (0.333; 0.698)	8 (19%)	20 (47.6%)	0.005 (−0.478; −0.093)	3 (7.1%)	9 (21.4%)	0.060 (−0.289; 0.003)	0	4 (9.5%)	NA

**Table 2 vaccines-12-00107-t002:** Unvaccinated children (percentage and number) by reasons for non-vaccination in urban and rural settings in Romania.

Reasons for Non-Vaccination	Urban Settings (N = 3610)	Rural Settings (N = 2831)	*p*
Failure to attend the GP’s office	45.9% (N = 1658)	37.9% (N = 1072)	0.001
Temporary medical contraindications	25.9% (N = 934)	25.5% (N = 723)	0.850
MMR vaccination refusal	9.9% (N = 357)	7.8% (N = 220)	0.610
Children lost to follow-up (born/living abroad)	7.6% (N = 276)	11.7% (N = 330)	0.150
Vaccine unavailable	6.4% (N = 232)	8.7% (N = 245)	0.393

**Table 3 vaccines-12-00107-t003:** Measles vaccine coverage rates for the 3 most affected counties during the 2023 measles outbreak in Romania.

County	MMR1 (%) 2022	Number of Eligible Children Aged 12 Months	MMR2 (%) 2022	Number of Eligible Children Aged 5 Years
National average	83.4		71.4	
Mures	72.9	4665	68.7	6223
Brasov	76.4	5175	59.4	6736
Cluj	87.9	6399	73.4	7524
Bucharest	84	15,555	70	22,240

MMR1—measles–mumps–rubella vaccine, first dose—administered at age 12 months (birth cohort 2021); MMR2—measles–mumps–rubella vaccine, second dose administered at age 5 years, (birth cohort 2017).

## Data Availability

All data presented are available upon request from the first author (A.S.).

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
