# Peer review of "Suboptimal MMR Vaccination Coverages—A Constant Challenge for Measles Elimination in Romania"

_vaccines, 2024, doi:10.3390/vaccines12010107_

Round 1
Reviewer 1 Report
Comments and Suggestions for Authors
This paper is a "cry of alarm" regarding the global treath of sub-optimal measles vaccination. The scenario, which would cause the regress of the world, can be avoided if the Health Authorities at National, European and Global level, give the right support of human, logistic, economic and organizational resources to the relaunch of vaccination activities. Otherwhise the Middle age and its darkness will retutn. This paper must reach the tables of political decision makers in Romania, so that in the future it will not be said that no one has raised the alarm. The paper can be published promptly also becuse it describes the risk that could come from war scenarios that are bloodying Europe and the World.
Author Response
Response to Reviewer 1
This paper is a "cry of alarm" regarding the global treath of sub-optimal measles vaccination. The scenario, which would cause the regress of the world, can be avoided if the Health Authorities at National, European and Global level, give the right support of human, logistic, economic and organizational resources to the relaunch of vaccination activities. Otherwhise the Middle age and its darkness will retutn. This paper must reach the tables of political decision makers in Romania, so that in the future it will not be said that no one has raised the alarm. The paper can be published promptly also becuse it describes the risk that could come from war scenarios that are bloodying Europe and the World.
We are grateful for the positive recommendation; the article is indeed an alarm signal for an urgent public health matter.
Reviewer 2 Report
Comments and Suggestions for Authors
The manuscript "Suboptimal MMR Vaccination Coverages – A Constant Challenge for Measles Elimination in Romania" by Aurora Stanescu et al. appears to be a relevant contribution to the field of epidemiology and infectious diseases, particularly concerning measles vaccination efforts in Romania. However, there are key areas where the manuscript could be significantly improved:
Depth of Analysis: The work primarily analyzes vaccination levels and compares them with morbidity rates. This approach lacks a deeper, more insightful analysis, such as investigating seropositivity indicators.
Comparative Perspective: The absence of a comparative analysis with other countries, either within the region or with similar socioeconomic standards, is a notable omission. Such a comparison could provide valuable context and enhance the significance of the findings.
Originality and Content Use: The section on reasons for non-vaccination (3.4) heavily relies on previously conducted studies without proper referencing. This not only raises concerns about originality but also the effective use of content within the article.
Given these considerations, the manuscript does not yet meet the standards of a full-fledged research article. It could be repositioned as a short communication after thorough revision, particularly by excluding previously published materials and enhancing original analysis.
Author Response
Response for Reviewer 2

Reviewer 3 Report
Comments and Suggestions for Authors
This study analyzes the consequences of inadequate measles vaccination coverage in Romania. As expected, Romania has experienced two major measles epidemics in the last two decades. The study focuses on the outbreak that started in 2020 and is still ongoing, causing over 1,000 confirmed cases per year, most of which occur in unvaccinated children aged 1-4 years. The study is well-written and easy to understand, but offers little, if anything, new. There is no description of the clinical consequences of infection, viral load and transmission rate of unvaccinated or vaccinated children with one or two doses. Finally, the circulating genotype was determined by a different group and in a different study. The article discusses the consequences of low vaccination coverage and the risks of infection due to heavy immigration pressure from Ukraine and the ongoing conflict. This article is more appropriate for a public health journal.
Author Response
Response to Reviewer 3

Reviewer 4 Report
Comments and Suggestions for Authors
The paper is interesting and enlightening about issues related to MMR vaccination in Romania. It is easy to read and is well written. To improve the paper, the following issues should be resolved.
MAJOR ISSUES
1. Authors should use a joinpoint regression to study the trends in vaccination. You can use the joint point
2. In Table 1, compute a chi-square test for 12-month-old children and another for 18-month children.
3. Table 1 should have the total. All the categories should be exhaustive, eg <75%, 75-90%, 90-95, > 95%.
4. In the results line 160, there is the text “a communities, (45.9 % in urban vs 37.9% in rural, p = 0.001) “ , but nowhere is explained what statistical test was computed, Z, chi-square, Fisher, etc.
5. In the material and methods section, the author should explain the statistical test they used and the software.
Minor issues
6. Please use the same terminology in the paper. In Figure 1, MMR1 and MMR2 are used. Meanwhile, in the text, MCV1 and MCV2 are used.
7. For the discussion, explain what happened in 2014 to justify the abrupt decrease.
Author Response
Response to Reviewer 4

Reviewer 5 Report
Comments and Suggestions for Authors
Thank you for sharing your manuscript investigating MMR vaccination coverage in Romania. The following comments may help to improve the article.
L33: Please be more specific concerning the vaccine to be administered "in all population groups" as well as "in all geographic areas".
L45: Please include a statement at what age children should receive the first and the second dose of the MMR vaccine.
L73: In which parts of Romania did the outbreaks/epidemic occur?
L82: Was rubella also reported during the second measles outbreak in 2016-2020?
L92: Please include in the background section some information on the measles/MMR vaccination in Romania. I think this is pivotal to better understand vaccine coverage rates and issues like vaccine hesitancy.
L106-109: Who collects and fills in the data gathered by the questionnaire? At what time points throughout the year are these data collected and what is the reporting structure in place? How are data verified for completeness and correctness?
L149-150: Please add an up-to-date suitable reference for the statement that 95% coverage is required for interrupting measles transmission. What about herd immunity?
Table 1: Do you mean 21,4% or 21.4% in the category VCR <75% 18mo in 2022.
L157: Please include the questionnaire at least as a supplementary material in your manuscript to better understand the data assessed.
L162: What plausible reasons could exist to explain vaccination refusal likely among caretakers/parents?
L170: How are measles cases confirmed in Romania?
L179: What other preventive measures could be implemented for children <1 year of age who are not eligible for MCV vaccination?
L183-184: What could be the rationale behind the spread seen?
L192-193: In which way could the population and its transmission patterns vary?
L205: Migration does not necessarily prevent vaccination. Is the Romanian population migrating or other nationalities migrating to Romania? Please clarify.
L 206-207: How is the perception, the trust in general practitioners?
L209: What do you mean by temporary contradictors? Please be more specific as temporary issues could likely be addressed.
L204-211: Are those finings from your investigation or are you referring to reference(s); please clarify. If you are citing from reference(s), please include them here.
Comments on the Quality of English Language
Please see above.
Author Response
Response for Reviewer 5

Round 2
Reviewer 2 Report
Comments and Suggestions for Authors
The author's response to the reviewer's comments indicates that they have made significant revisions to the manuscript, including addressing the lack of a comparative analysis with other countries, clarifying the originality of the content, and providing additional information on seroprevalence and vaccination coverage rates in the WHO European Region and the European Union. The author also acknowledges the need for a deeper analysis and states that the manuscript could be repositioned as a short communication.
Based on the provided response, the manuscript appears to have been substantially revised to address the reviewer's comments. The addition of comparative analysis and original data on vaccination coverage rates and measles cases in the European region enhances the significance of the findings. The clarification of the originality of the content and the exclusion of previously published materials also strengthen the manuscript.
Author Response
We are grateful for the constructive review; we consider that the comments and suggestions had helped improved the quality of our article.
Reviewer 4 Report
Comments and Suggestions for Authors
See attached document

Author Response
We are grateful for the constructive review; we consider that the comments and suggestions had helped improved the quality of our article. We tried to clarify the following point:
Reviewer 4
Dear Authors
Thank for introducing all our sugestions in the manuscript. The paper have improved a lot. Nevertheless an important question remains to be made.To do a joinpoint regression with the data of figure 1. Vaccination coverage with the first and the second dose of MMR vaccine in Romania, 2005- 2022.
A sentence was added in the manuscript at lines 125 – 132.
Following reviewer 4’s recommendations we performed joinpoint regression with the data of figure 1. Vaccination coverage with the first and the second dose of MMR vaccine in Romania, 2005- 2022 available in manuscript at lines 137 – 158.
Reviewer 5 Report
Comments and Suggestions for Authors
Thank you for sharing the revised manuscript and addressing all my comments.
Table 1/Table 2: Please check the journal's guidelines how to report p-values and 95% CIs, i.e., the number of decimal places. Also, any abbreviations used should be explained next to a table/figure.
Comments on the Quality of English LanguagePlease see above.
Author Response
We are grateful for the constructive review; we consider that the comments and suggestions had helped improved the quality of our article. We tried to clarify all the points as follows:
Reviewer 5
Thank you for sharing the revised manuscript and addressing all my comments.
Table 1/Table 2: Please check the journal's guidelines how to report p-values and 95% CIs, i.e., the number of decimal places. Also, any abbreviations used should be explained next to a table/figure.
Corrections have been done for Table 1 in manuscript at line 200 and for Table 2 in manuscript at line 217 – 220
We made corrections required for minor editing of English language in the manuscript.